# Gut Microbiota in Non-Alcoholic Fatty Liver Disease Patients with Inflammatory Bowel Diseases: A Complex Interplay

**DOI:** 10.3390/nu14245323

**Published:** 2022-12-15

**Authors:** Ludovico Abenavoli, Lidia Giubilei, Anna Caterina Procopio, Rocco Spagnuolo, Francesco Luzza, Luigi Boccuto, Emidio Scarpellini

**Affiliations:** 1Department of Health Sciences, University “Magna Graecia”, 88100 Catanzaro, Italy; 2School of Nursing, Clemson University, Clemson, SC 29634, USA; 3Translational Research in Gastrointestinal Disorders (T.A.R.G.I.D.), Gasthuisberg University Hospital, KU Leuven, Herestraat 49, 3000 Leuven, Belgium

**Keywords:** microbiota, non-alcoholic fatty liver disease, Crohn’s disease, ulcerative colitis

## Abstract

The intestinal microbiota represents the microbial community that colonizes the gastrointestinal tract and constitutes the most complex ecosystem present in nature. The main intestinal microbial phyla are *Firmicutes*, *Bacteroidetes*, *Actinobacteria*, *Proteobacteria*, *Fusobacteria*, and *Verrucromicrobia*, with a clear predominance of the two phyla *Firmicutes* and *Bacteroidetes* which account for about 90% of the intestinal phyla. Intestinal microbiota alteration, or dysbiosis, has been proven to be involved in the development of various syndromes, such as non-alcoholic fatty liver disease, Crohn’s disease, and ulcerative colitis. The present review underlines the most recurrent changes in the intestinal microbiota of patients with NAFLD, Crohn’s disease, and ulcerative colitis.

## 1. Introduction

The intestinal microbiota represents the microbial community colonizing the gastrointestinal (GI) tract. It has been estimated that the GI tract hosts a population greater than 100 trillion microorganisms, making it the most complex ecosystem present in nature [1]. The gut microbiota is composed of different species of microorganisms, amongst which we find eukarya, bacteria, archaea, and viruses. Undoubtedly, bacteria represent the main strains. The main intestinal bacterial phyla are represented by *Firmicutes*, *Bacteroidetes*, *Actinobacteria*, *Proteobacteria*, *Fusobacteria*, and *Verrucromicrobia*, with a clear predominance for the two phyla *Firmicutes* and *Bacteroidetes*, accounting for almost 90% of the entire population [2]. Microbiota begins colonizing the host at the moment of delivery, namely birth, although the paradigm of uterus sterility has recently been questioned [3]. During and after birth, the intestines of newborns are colonized by a series of microbes. This process is conditioned by various factors: mode of delivery, type of breastfeeding, the hygienic conditions, and exposure to antibiotic treatment. Usually, the intestinal microbial population assumes the configuration of an adult microbiota in the first five years of life, even if it represents an ecosystem with a dynamic evolution in the course of life [4]. The intestinal microbiota is defined as a superorganism essential for the health of the host. The microbiota fulfills various functions, such as immune homeostasis; it is essential in countering colonization by pathogenic bacteria and maintaining the integrity of the intestinal barrier. In addition, it is involved in the extraction, synthesis, and absorption of bile acids, vitamins, amino acids, and short-chain fatty acids (SCFAs) [2,5]. Several studies have shown that in conditions of equilibrium—namely “eubiosis”—gut microbiota is essential for the fulfillment of crucial functions for the host. On the other hand, in conditions of derangement—namely “dysbiosis”—gut microbiota is involved in the development of different pathologic conditions [6]. For example, we can find non-alcoholic fatty liver disease (NAFLD) and inflammatory bowel diseases (namely, Crohn’s disease and ulcerative colitis). NAFLD is the most common chronic liver disease worldwide and affects approximately 25% of the general population [7]. NAFLD is a clinical condition characterized mainly by the accumulation of fat in the hepatic parenchyma (>5% of hepatocytes) [8]. The acronym NAFLD is an umbrella term that encompasses a large spectrum of conditions, ranging from simple fatty liver to non-alcoholic steatohepatitis (NASH), which can evolve into liver cirrhosis and hepatocellular carcinoma [9]. NAFLD is a multifactorial disease whose development is linked to environmental, genetic, and dietary factors, metabolic unbalances such as insulin resistance, lipotoxicity, altered inflammatory response and immunity with micro-inflammation, cytokine imbalance, activation of innate immunity, and last, but not least, alterations in the microbiota.

Inflammatory bowel diseases (IBD) include two main forms: Crohn’s disease and ulcerative colitis, which are configured as chronic recurrent inflammatory bowel diseases [10,11]. Crohn’s disease is responsible for a transmural inflammation that can affect any area of the GI tract from the mouth to the anus in a non-continuous way, although it is mainly localized at the level of the terminal ileum or the perianal region. On the contrary, ulcerative colitis is responsible for inflammation of the mucosa, limited to the colon, a phenomenon that makes this syndrome less heterogeneous [12]. Due to its heterogeneity, Crohn’s disease presents signs and symptoms that vary according to the severity and location of the disease. Among the most frequently highlighted symptoms, we find colicky abdominal pain, anemia, watery diarrhea, fever, weight loss, and asthenia. On the contrary, in ulcerative colitis, bloody diarrhea is the most frequently encountered manifestation, accompanied by tenesmus, abdominal pain, anemia, weight loss, and fever [13]. Although the etiopathogenesis of IBD has not yet been completely clarified, it has now emerged that these diseases are the result of a complex interaction of genetic, environmental factors, alterations in the immune component, and intestinal dysbiosis.

Both diseases, either NALFD and IBD, recognize common pathogenetic actors: diet, genetic background, metabolism dysregulation, altered inflammatory and immune system functioning, altered intestinal permeability, and last, but not least, gut dysbiosis. The latter can be the first actor or conditioned by the aforementioned in a vicious circle.

Thus, the present review focused on underlining how the alterations of the microbiota can constitute a crucial element in the development of NAFLD, Crohn’s disease, and ulcerative colitis.

## 2. Gut Microbiota and NAFLD

NAFLD is a multifactorial disease where dysbiotic factors are emerging as crucial in its pathophysiology. In particular, several studies have analyzed the composition of the intestinal microbiota in NAFLD and healthy subjects observing a peculiar dysbiosis. In a prospective cross-sectional study conducted by Wang et al., the fecal microbiota of NAFLD patients was compared with those of healthy subjects [14]. During this study, 126 subjects were enrolled (43 were affected with NAFLD and 83 were healthy volunteers). The study results indicated that individuals with NAFLD presented higher fecal levels of the *Bacteroidetes* phylum vs. healthy subjects (66.0% vs. 46.4%, *p* = 0.005), and lower levels of the *Firmicutes* phylum (27.5% vs. 51.9%, *p* = 0.002). It was also found that the *Bacteroidia* class presented higher levels in patients with NAFLD (65.93% vs. 46.18%, *p* = 0.004), while the *Clostridia* class was significantly reduced in these patients when compared to healthy subjects (27.02% vs. 50.91%, *p* = 0.001). The characterization of the microbiota of subjects with NAFLD vs. controls also indicated a marked decrease in the *Lachnospiraceae* (18.67% vs. 34.26%, *p* = 0.002), *Ruminococcaceae* (6.69% vs. 13.67%, *p* = 0.018) *Lactobacillaceae* (0.04% vs. 0.73%, *p* = 0.0008) and *Peptostreptococcaceae* families (0.09% vs. 0.44%, *p* = 0.002). Furthermore, the fecal microbiota of subjects with NAFLD compared to those of healthy subjects showed a decrease in genera: *Coprococcus* (0.34% vs. 2.4%, *p* = 0.009), *Pseudobutyrivibrio* (0.45% vs. 2.16%, *p* = 0.02), *Moryella* (0.05% vs. 0.35%, *p* = 0.02), *Roseburia* (1.48% vs. 2.59%, *p* = 0.01), *Anaerosporobacter* (1.08% vs. 2.02%, *p* = 0.02), *Anaerotruncus* (0.19% vs. 0, 96%, *p* = 0.004) and *Ruminococcus* (1.03% vs. 1.59%, *p* = 0.02). In a study conducted by Tsai et al., 75 adult subjects were enrolled, including 25 with NAFLD, 25 with NASH, and 25 with no health issues [15]. During the study, the taxonomic composition of the gut microbiota was obtained by 16S ribosomal RNA gene sequencing from stool samples. The abundance of the *Lentisphaerae* phylum was significantly lower in NAFLD and NASH patients than in healthy subjects (*p* < 0.01). Moreover, the *Clostridia* class had significantly lower levels in subjects with NAFLD than in controls (*p* < 0.05). Furthermore, the NAFLD group was characterized by a reduction in the levels of the *Clostridiales* order (*p* < 0.05) and of the *Ruminococcaceae* family (*p* < 0.05) vs. the healthy subjects.

In a study by Jiang et al., differences between the gut microbiota of NAFLD and healthy subjects were analyzed [16]. In total, 85 patients were enrolled: 53 were affected with NAFLD and 32 were healthy volunteers. The diagnosis of NAFLD was made through ultrasound or histological examinations. The predominant phyla in these patients were *Bacteridetes* and *Firmicutes*. Of the remaining phyla, only the data referring to the *Lentisphaerae* phylum showed statistically significant differences. In particular, the NAFLD group presented a significant decrease in *Lentisphaerae* vs. the healthy group (*p* < 0.05). In addition, the genus *Clostridium XI* showed an approximately 2-fold increase in the NAFLD group compared to the healthy group (*p* < 0.01). Similar findings were also observed for the *Lactobacillus* genus, found to be higher in NAFLD subjects (*p* < 0.05). It was also observed that the genera *Oscillibacter* and *Flavonifractor*, belonging to the *Ruminococcaceae* family, had significantly reduced levels in the NAFLD group vs. the healthy subjects group (*p* <0.05). The genera *Odoribacter* (belonging to the *Bacteroidetes* phylum), and the *Allistipes* genus (belonging to the *Rikenellaceae* family) showed a statistically significant decrease in their levels in the NAFLD vs. healthy subjects (*p* <0.01).

In a study by Lino et al., the fecal microbiota of a large cohort of subjects was examined [17]. The relative abundance of different taxa of fecal samples was examined using 16 S ribosomal RNA amplification. The diagnosis of NAFLD was made by abdominal ultrasound. Of the 1148 patients initially enrolled, fecal samples were collected from 874: 669 patients composed the healthy group and 205 patients the NAFLD group. There was a statistically significant decrease in the Clostridia class in the NAFLD group vs. healthy subjects’ group (*p* = 0.015). A similar finding was observed in the case of the *Clostridiales* (*p* = 0.015) and *Selenomonadales* orders (*p* = 0.004), the *Ruminococcaceae* (*p* = 0.007), *Bacteroidaceae* (*p* = 0.032), and *Veillonellaceae* families (*p* = 0.024), and for the *Bacteroides* (*p* = 0.024), *Faecalibacterium* (*p* = 0.025), and *Lachnospiracea_incertae_sedis* genera (*p* = 0.031).

## 3. Gut Microbiota and IBD

Different studies observed changes in the gut microbiota of IBD patients. In a study by Fujimoto et al., a cohort of patients including 47 subjects with Crohn’s disease and 20 healthy controls was enrolled [18]. Interestingly, the abundance of *Faecalibacterium prausnitzii*, quantified by a real-time polymerase chain reaction, was significantly reduced in subjects with Crohn’s disease vs. healthy subjects (*p* = 0.0004). In a study conducted by Takahashi et al., fecal samples from 10 patients with inactive Crohn’s disease and 10 healthy individuals underwent 16S rRNA sequencing [19]. Butyrate-producing bacterial species, such as *Blautia faecis*, *Roseburia inulinivorans*, *Ruminococcus torques*, *Clostridium lavalense*, *Bacteroides uniformis*, and *Faecalibacterium prausnitzii*, recorded a significant reduction vs. in healthy individuals (*p* < 0.05).

In a study by Ma et al., changes in the fecal microbiota of patients with inflammatory bowel disease were analyzed [20]. During the study, fecal samples were collected from 15 patients with Crohn’s disease, 14 patients with ulcerative colitis, and 13 healthy subjects, respectively. The collected samples were analyzed using 16S ribosomal DNA gene sequencing. Characteristically, the diversity and microbial structure of patients with Crohn’s disease and ulcerative colitis were significantly different from healthy controls. In particular, the relative abundance of *Bacteroidetes* in both patient groups was decreased vs. healthy subjects (Crohn’s disease vs. control, 47.49%, and 66.85%, respectively, *p* = 0.015; ulcerative colitis vs. control, 48.94% and 66.85%, respectively, *p* = 0.019). On the other hand, proteobacteria showed significantly increased levels in both Crohn’s disease and Ulcerative colitis groups vs. healthy controls (Crohn’s disease vs. control, 26.79% and 7.34%, respectively, *p* = 0.002; Ulcerative colitis vs. control, 17.48% and 7.34%, respectively, *p* = 0.005). In a study by Andoh et al. [21], 31 patients with ulcerative colitis, 31 with Crohn’s disease, and 30 healthy individuals were enrolled for fecal microbiota composition study. The 16S rRNA genes of the fecal samples were amplified by the polymerase chain reaction. Typically, patients suffering from Crohn’s disease and ulcerative colitis had a significant decrease in the *Clostridium* family. Of mention, no significant differences were observed between patients with active ulcerative colitis and those with active Crohn’s disease. However, this evidence is not uniform. In a study by Shah et al., colon biopsy samples from 10 patients with ulcerative colitis and 13 healthy controls were analyzed [22]. A 16S rRNA gene sequencing of V4–V6 variable regions was performed using the Illumina MiSeq instrument. At the phylum level composition, the two groups were similar with the exception of the reduction in the *Verrucomicrobia* phylum in patients with ulcerative colitis vs. healthy controls (*p* < 0.02). At the genus level, a significant reduction in *Roseburia* was observed in patients with ulcerative colitis only (*p* = 0.03).

## 4. NAFLD in IBD: An Unexplored Connecting Gut-Liver-Immune Axis

Despite the lack of a well defined and characterized common pathogenesis, the presence of a possible physiopathologic interconnection between NAFLD and IBD is emerging, according to various scientific evidence. First, there a statistical association between NAFLD prevalence and IBD. A meta-analysis of 27 studies indicated that the prevalence of NAFLD in patients with IBD was 32% (95% CI, 24–40%), significantly higher than in the general population (~25%) [23]. In a retrospective study including 694 patients with IBD, it was observed that 48% of patients with Crohn’s disease and 44% of patients with ulcerative colitis suffered from NAFLD [24]. In a study by Principi et al. including 465 subjects with IBD and 189 healthy subjects, it was observed that NAFLD had rates of 28.0% in IBD subjects and 20.1% in healthy subjects (*p* = 0.04), respectively [25].

Several hypotheses have been advanced in an attempt to understand the possible common pathophysiological basis of both syndromes Although there is theory indicating that genetic susceptibility is one of the crucial factors for the development of liver steatosis in IBD, changes in the intestinal microbiota associated with the impairment of permeability of the intestinal barrier are a solid and growing point of evidence (Figure 1) [26].

The intestinal barrier constitutes a physical barrier dedicated to the defense of the organism. This barrier represents an architecture finely orchestrated by different components, such as the adhesive mucous gel layer, immunoglobulins, antibacterial peptides, and the tight intercellular junctions. Among these components, the tight junctions constitute a true pillar, whose particularly complex structure is constituted by transmembrane proteins (namely, claudin and occluding), junctional adhesion molecule-A (JAM-A), and proteins of the intracellular plaque, such as zonula occludens [27]. In detail, an analysis of biopsy samples of the sigmoid colon of patients with Crohn’s disease showed an under-regulation of the occludin protein and the sealing proteins of the tight junction claudin 5 and claudin 8. Moreover, there was an over-regulation of the pore-forming protein claudin 2 [28]. Analysis of colorectal mucosal tissue samples from IBD patients revealed a dramatic and global down-regulation of the transmembrane protein occludin [29].

The increased permeability of the intestinal barrier with the increased exposure of immune cells to microbial-associated molecular patterns (PAMP) leads to an induction of the inflammatory response, adaptive immunity, and activation of various types of T lymphocytes (Th1, Th2, Th9, Th17, and Treg) [30]. In this frame, genetic predisposition is another piece of the pathophysiological puzzle of IBD with NALFD. In particular, genetic polymorphisms associated with the expression of proteins for altered tight junctions are correlated with an increased risk of developing IBD. Among these, we find those involving genes encoding tight junction-associated proteins, such as myosin IXB (*MYO9B*), membrane-associated inverted guanylate kinase 2 (*MAGI2*), proteins linked to cell adhesion such as E-cadherin (*CDH1*), and proteins of the mucus layer such as mucin 3A and 19 [31].

On the side of NAFLD, increased intestinal permeability, induced by the alteration of the microbiota whose causative factors are diet, altered metabolism (obesity, diabetes), and systemic micro-inflammation, is also a crucial factor in the development of NAFLD. In fact, increased intestinal permeability determines a similar passage of the PAMPs, that through the systemic portal circulation, finds the first organ, landing in the liver. Several pieces of evidence suggest that in the liver the lipopolysaccharide, a PAMP located in the external wall of Gram-negative bacteria, interacts with the toll-like receptors (TLR) [32]. TLR4, highly expressed in Kupffer cells, following the activation determined by the LPS, will lead to the activation of nuclear transcription factors such as NF-κB, triggering a pro-inflammatory cascade typical of NAFLD [33]. In this regard, the promotion of the transcription of NF-κB represents the first signal for the activation of the inflammasome. Inflammasomes consist of cytosolic multiprotein oligomers belonging to the innate immune system. Thus, we can consider it a “sensor protein” (namely, an adapter protein, and a caspase-1). Examples of inflammosome include NOD-like receptors (NLRs), such as NLRP1, NLRP3, and NLRP6 [34]. The activation of the inflammasome requires stimulation by two different signals. The first is represented by the interaction of PAMPs with TLRs resulting in the activation of NF-κB [35,36]. The latter, in turn, stimulates the expression of the NLRP3 inflammasome components, proIL-1β and proIL-18, which after translocation from the nucleus to the cytoplasm, remain inactive until the presence of the second signal [34,37]. The second activation signal of the inflammasome is characterized by different mechanisms, including the entry of PAMPs into the cell through the pannexin-1 channels, which are opened following the interaction of ATP with the purinergic receptor P2×7. The second signal can determine the oligomerization of the inactive complex of the NLRP3 inflammasome, resulting in maturation and up-regulation of IL-1β and IL-18 [38]. In this context, it is interesting to observe that the activation of the inflammasome NLRP3 has been associated with the development of IBD and the progression from NAFLD to NASH [39,40]. Thus, altered gut microbiota and impaired intestinal permeability start an inflammatory cascade in both NAFLD and IBD.

Second, the metabolites produced by the microbiota also play a crucial role in the pathophysiology of both NAFLD and IBD and their physiopathologic connection. Among these we find SCFA, branch chain amino acids (BCAA) and bile acids.

In detail, SCFAs are linear carboxylic acids characterized by a carbon chain with less than six carbon atoms (C < 6). The most important SCFAs are represented by acetic acid, propionic acid and butyric acid, characterized respectively by two, three and four carbonaceous units. The production of SCFA occurs in the intestine starting from the dietary fiber typical of plant foods [41]. Dietary fibers can travel intact through the upper digestive tract to the cecum and to the large intestine, and be digested by gut microbiota. The latter provide enzymatic pathways that the host lacks. SCFAs perform crucial activities for maintenance of the intestinal barrier and regulation of the host’s immune tolerance. Different microbial compositions produce different amounts and types of SCFAs [42,43]. In this regard, *Bacteroidetes* are mainly associated with the production of acetate and propionate, while *Firmicutes* are associated with the production of butyrate. Thus, gut dysbiosis can be associated with increased intestinal permeability, initiation of altered gut-liver axis and initiation of both NAFLD and IBD [44].

In detail, in NAFLD subjects, butyrate-producing probiotics reduce the accumulation of lipids in the liver by improving hepatic resistance to insulin through the activation of AMP-activated protein kinase (AMPK), and the expression of factor-related erythroid nuclear factor 2 2 (Nrf2) in rats with NAFLD. Additionally, SCFA supplementation is associated with a reduction in hepatic fat deposition, decreased fatty acid synthase activity, increased lipid oxidation, and suppressed expression of the inflammatory cytokines IL-6 and the TNFα [45].

In IBD patients, SCFAs appear to play an anti-inflammatory role. In a study by Chen et al., the anti-inflammatory activity of sodium butyrate was evaluated on mice with colitis induced by 2,4,6-trinitrobenzene sulfonic acid (TNBS), an experimental model of Crohn’s disease [46]. Sodium butyrate significantly improved the inflammatory response and intestinal epithelial barrier dysfunction by activating GPR109A, and inhibiting LPS-induced phosphorylation of NF-κB p65 and AKT signaling pathways. In a study by Zhang et al., the effect of butyrate in a model of rat colitis induced by 2,4,6-trinitrobenzene sulfonic acid was evaluated [47]. Butyrate administration suppressed IL-17 levels in both plasma and colonic mucosa, and improved colitis lesions of the colon.

Bile acids also play a key role in the development of both NAFLD and IBD. They are produced in the liver through a process of oxidation of cholesterol; they can be divided into primary bile acids (e.g., cholic acid and chenodeoxycholic acid), conjugated bile acids and secondary bile acids (e.g., deoxycholic acid and lytic acid). Under physiological conditions, these compounds allow the absorption of fatty acids, promote the metabolism of glucose through the activation of the farnesoid X receptor (FXR) and the G protein-coupled bile acid receptor (TGR5) [48]. Thus, bile acids are the most representative molecules of the gut–liver axis. In fact, after being synthesized in the liver, bile acids are poured into the duodenum through the bile, converted into secondary bile acids through the microbiota, and reabsorbed after having performed their functions at the level of the distal ileum to be transported to the liver through the portal circle [49]. Moreover, several studies have shown that gut dysbiosis is associated with changes in the composition of the bile acid pool, resulting in the reduction in the FXR and TGR5 signaling and the further development of NAFLD. In more detail, in NAFLD patients there is an increase in plasma levels of bile acids and a reduction in hepatic FXR levels, with the greater expression of SREBP-1C and hepatic triglyceride synthesis [50]. Furthermore, in a study conducted by Jiao et al., in children with NAFLD, serum concentrations of primary and secondary bile acids and concentrations of intestinal bile acid-producing bacteria are increased. On the contrary, hepatic signaling of bile acids mediated from FXR is suppressed [51]. Finally, the analysis of bile acid synthesis intermediates also seems to confirm an increase in plasma levels of bile acids in NAFLD subjects (namely, of 7α-hydroxy-4-cholesten-3-one) [52,53].

In the frame of common physiopathologic liaison between NAFLD and IBD, bile acids also appear to be implicated in the development of IBD. Gut microbiota dysbiosis is associated with a drastic reduction in the levels of secondary bile acids in these patients. In fact, there is a similar lack of activation of FXR and TGR5 receptors by bile acids, responsible for the break-down of the integrity of the intestinal barrier with an increase in bacterial translocation. Further, it has been hypothesized that the reduction in secondary bile acids could lead to a disruption of the balance between Th17 and Treg cells in IBD patients [54].

Gut dysbiosis and the alteration of the intestinal signaling typical of the IBD framework, acting through factors such as GI hormones and peptides, are also connected to the control of satiety/hunger and bile acids metabolism. Altogether, these may be responsible for the onset of obesity (e.g., increased food intake, increased high-fat diet ingestion), dysmetabolism (namely, liver and systemic fat deposition, diabetes onset), and, finally NAFLD [55]. In this regard, in a study conducted by Shen et al., a model of dextran sulfate sodium salt (DSS)-induced colitis was created by evaluating how intestinal dysbiosis was associated with the development of NASH [56]. DSS-induced colitis can promote the progression of inflammation and liver fibrosis by inducing microbiota dysbiosis, resulting in the triggering of an inflammatory response with the disruption of angiocrine signaling in non-parenchymal liver cells. On the other hand, IBD patients present with typical malnutrition, a condition closely linked to a vitamin deficiency, a factor involved in the development of liver pathologies [57].

## 5. Conclusions

Currently, IBD and NAFLD represent major global public health problems. The establishment of comorbidity between the two syndromes is an increasingly common phenomenon that causes an enormous economic and healthcare burden. In this context, the identification of the factors associated with the development of both syndromes and with the identification of the common pathophysiological path can represent significant progress in terms of diagnostic and therapeutic strategy. Thus, the study, analysis, and expansion of knowledge regarding the alterations of the microbiota, that distinguish and link together both syndromes, represent the principal path for future diagnostic and therapeutic approaches.

## Figures and Tables

**Figure 1 nutrients-14-05323-f001:**
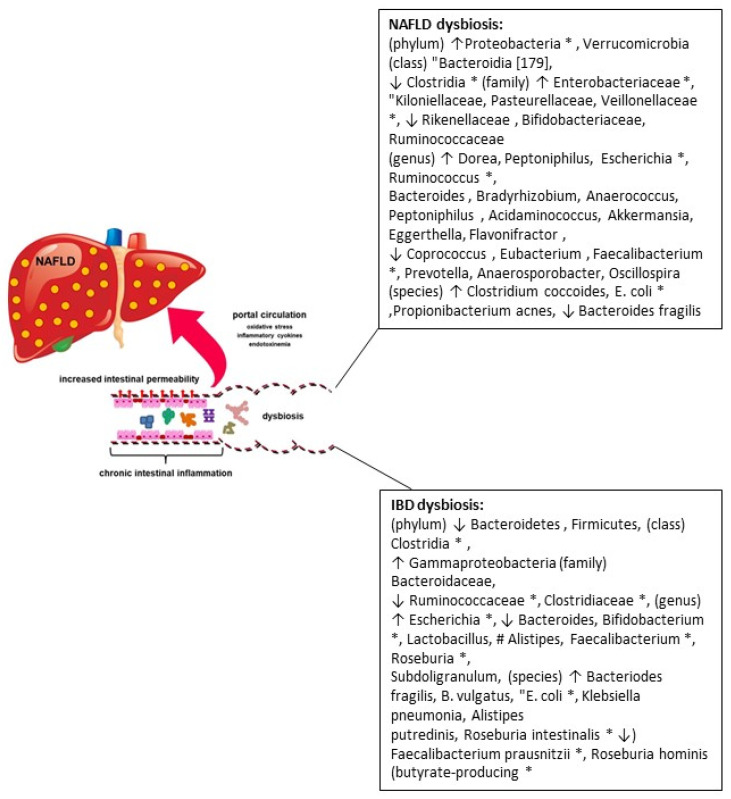
Increase in intestinal permeability, oxidative stress (e.g., starting of inflammatory response with intestinal mucosa damage), endotoxemia (namely, lipopolysaccharide concentration) and gut dysbiosis in IBD patients, involved in the onset and development of NAFLD. ↑ increased abundance; ↓ decreased abundance; * shared gut dysbiosis between NAFLD and IBD patients.

## Data Availability

Data from the present review article can be retrieved from PudMed, MEDLINE, main Gastroenterological meetings database available online.

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
