# Peer review of "Gut Microbiota in Non-Alcoholic Fatty Liver Disease Patients with Inflammatory Bowel Diseases: A Complex Interplay"

_nutrients, 2022, doi:10.3390/nu14245323_

Round 1

Reviewer 1 Report

Dear Authors,

Interesting publication, but I am asking the authors to explain what is new in their review?

The layout of the paper is correct, but I don't think it is exhaustive, especially the section on inflammatory bowel disease and NAFLD. Maybe focus more specifically on the liver-gut axis?

The issue of the role of the microbiome and NAFLD is well covered in the following publications, which I believe would be useful to include in the review (for consideration by authors):

Jiang X, Zheng J, Zhang S, Wang B, Wu C, Guo X. Advances in the Involvement of Gut Microbiota in Pathophysiology of NAFLD. Front Med (Lausanne). 2020 Jul 29;7:361. doi: 10.3389/fmed.2020.00361. PMID: 32850884; PMCID: PMC7403443.

Tokuhara D. Role of the Gut Microbiota in Regulating Non-alcoholic Fatty Liver Disease in Children and Adolescents. Front Nutr. 2021 Jun 25;8:700058. doi: 10.3389/fnut.2021.700058. PMID: 34250000; PMCID: PMC8267179.

Zhou J, Tripathi M, Sinha RA, Singh BK, Yen PM. Gut microbiota and their metabolites in the progression of non-alcoholic fatty liver disease. Hepatoma Res. 2021 Jan 13;7:11. doi: 10.20517/2394-5079.2020.134. PMID: 33490737; PMCID: PMC7116620.

Vallianou N, Christodoulatos GS, Karampela I, Tsilingiris D, Magkos F, Stratigou T, Kounatidis D, Dalamaga M. Understanding the Role of the Gut Microbiome and Microbial Metabolites in Non-Alcoholic Fatty Liver Disease: Current Evidence and Perspectives. Biomolecules. 2021 Dec 31;12(1):56. doi: 10.3390/biom12010056. PMID: 35053205; PMCID: PMC8774162.

The relationship between IBD and NAFL is described in these sample publications, but I did not find them in the presented literature review.

Gibiino G, Sartini A, Gitto S, Binda C, Sbrancia M, Coluccio C, Sambri V, Fabbri C. The Other Side of Malnutrition in Inflammatory Bowel Disease (IBD): Non-Alcoholic Fatty Liver Disease. Nutrients. 2021 Aug 13;13(8):2772. doi: 10.3390/nu13082772. PMID: 34444932; PMCID: PMC8398715.

Shen B, Wang J, Guo Y, Gu T, Shen Z, Zhou C, Li B, Xu X, Li F, Zhang Q, Cai X, Dong H, Lu L. Dextran Sulfate Sodium Salt-Induced Colitis Aggravates Gut Microbiota Dysbiosis and Liver Injury in Mice With Non-alcoholic Steatohepatitis. Front Microbiol. 2021 Nov 2;12:756299. doi: 10.3389/fmicb.2021.756299. PMID: 34795650; PMCID: PMC8593467.

Augustyn M, Grys I, Kukla M. Small intestinal bacterial overgrowth and nonalcoholic fatty liver disease. Clin Exp Hepatol. 2019 Mar;5(1):1-10. doi: 10.5114/ceh.2019.83151. Epub 2019 Feb 20. PMID: 30915401; PMCID: PMC6431096.

Author Response

REPLY POINT BY POINT TO REVIWERS:

Changes/corrections are marked in red within the text.

REVIEWER 1:

Dear Authors,

Interesting publication, but I am asking the authors to explain what is new in their review?

The layout of the paper is correct, but I don't think it is exhaustive, especially the section on inflammatory bowel disease and NAFLD. Maybe focus more specifically on the liver-gut axis?

We thank the reviewer for these comments. We have revised and improved and focused the part on gut-liver axis. Moreover, we have adapted the title.

The issue of the role of the microbiome and NAFLD is well covered in the following publications, which I believe would be useful to include in the review (for consideration by authors):

Jiang X, Zheng J, Zhang S, Wang B, Wu C, Guo X. Advances in the Involvement of Gut Microbiota in Pathophysiology of NAFLD. Front Med (Lausanne). 2020 Jul 29;7:361. doi: 10.3389/fmed.2020.00361. PMID: 32850884; PMCID: PMC7403443.

Tokuhara D. Role of the Gut Microbiota in Regulating Non-alcoholic Fatty Liver Disease in Children and Adolescents. Front Nutr. 2021 Jun 25;8:700058. doi: 10.3389/fnut.2021.700058. PMID: 34250000; PMCID: PMC8267179.

Zhou J, Tripathi M, Sinha RA, Singh BK, Yen PM. Gut microbiota and their metabolites in the progression of non-alcoholic fatty liver disease. Hepatoma Res. 2021 Jan 13;7:11. doi: 10.20517/2394-5079.2020.134. PMID: 33490737; PMCID: PMC7116620.

Vallianou N, Christodoulatos GS, Karampela I, Tsilingiris D, Magkos F, Stratigou T, Kounatidis D, Dalamaga M. Understanding the Role of the Gut Microbiome and Microbial Metabolites in Non-Alcoholic Fatty Liver Disease: Current Evidence and Perspectives. Biomolecules. 2021 Dec 31;12(1):56. doi: 10.3390/biom12010056. PMID: 35053205; PMCID: PMC8774162.

We thank the reviewer for these suggestions. We have added them.

The relationship between IBD and NAFL is described in these sample publications, but I did not find them in the presented literature review.

Gibiino G, Sartini A, Gitto S, Binda C, Sbrancia M, Coluccio C, Sambri V, Fabbri C. The Other Side of Malnutrition in Inflammatory Bowel Disease (IBD): Non-Alcoholic Fatty Liver Disease. Nutrients. 2021 Aug 13;13(8):2772. doi: 10.3390/nu13082772. PMID: 34444932; PMCID: PMC8398715.

Shen B, Wang J, Guo Y, Gu T, Shen Z, Zhou C, Li B, Xu X, Li F, Zhang Q, Cai X, Dong H, Lu L. Dextran Sulfate Sodium Salt-Induced Colitis Aggravates Gut Microbiota Dysbiosis and Liver Injury in Mice With Non-alcoholic Steatohepatitis. Front Microbiol. 2021 Nov 2;12:756299. doi: 10.3389/fmicb.2021.756299. PMID: 34795650; PMCID: PMC8593467.

Augustyn M, Grys I, Kukla M. Small intestinal bacterial overgrowth and nonalcoholic fatty liver disease. Clin Exp Hepatol. 2019 Mar;5(1):1-10. doi: 10.5114/ceh.2019.83151. Epub 2019 Feb 20. PMID: 30915401; PMCID: PMC6431096.

We thank the reviewer for these suggestions. We have added them.

Reviewer 2 Report

The reviewed entitled” Gut microbiota in non-alcoholic fatty liver disease patients with inflammatory bowel disease “narrated microbiome signatures in NAFLD, IBD diseases including Chron’s disease and Ulcerative colitis. The authors also provided information on the role of SCFAs in NAFLD and IBD diseases. Finally, they also shed light on the interlink between NAFLD and IBD in the context of contribution of intestinal dysbiosis on the development of NAFLD. Overall, although the concept of review is pretty interesting the authors fails to provide enough information to support the title. Addressing the following comment would greatly help the manuscript.  

1.    First and foremost, although the review covered microbial signature changes in NAFLD and IBD individually, there is no information on NAFLD with IBD patients, to justify the title.

2.    Second, what kind of microbial changes results in NAFLD development and how? Authors provided link through SCFAs and inflammation is not new and not specific.

3.    Figure is not giving any new information and in fact authors never spelled the terms such as oxidative stress and endotoxemia in the content. The authors should list the microbial signatures in IBD and show their contribution to NAFLD development, specifically, rather general.

4.    Please provide a table or illustration showing the microbiota signatures in NAFLD, IBD and with combination.

5.    The authors described the mechanisms involved in the development of IBD with changes in microbiota, similar manner they should provide some information on what pathway they affect within the liver.

6.    How SCFAs contribute to NAFLD development under IBD? The review touched a sentence on the role of SCFAs on lipogenesis on adipose tissue (Line 195-196), but this is not sufficient and does not makes sense.

Again, overall the link between the NAFLD and IBD is missing and therefore provide more information to make the review interesting. 

Author Response

REPLY POINT BY POINT TO REVIWERS:

Changes/corrections are marked in red within the text.

REVIEWER 2.

The reviewed entitled” Gut microbiota in non-alcoholic fatty liver disease patients with inflammatory bowel disease “narrated microbiome signatures in NAFLD, IBD diseases including Chron’s disease and Ulcerative colitis. The authors also provided information on the role of SCFAs in NAFLD and IBD diseases. Finally, they also shed light on the interlink between NAFLD and IBD in the context of contribution of intestinal dysbiosis on the development of NAFLD. Overall, although the concept of review is pretty interesting the authors fails to provide enough information to support the title. Addressing the following comment would greatly help the manuscript.  

We thank the reviewer for all the observations. We have updated the manuscript and adapted the title in order to make it more specific and appropriate.

  1. First and foremost, although the review covered microbial signature changes in NAFLD and IBD individually, there is no information on NAFLD with IBD patients, to justify the title.

We thank the reviewer for this suggestion. We have provided the required information within Figure 1.

  1. Second, what kind of microbial changes results in NAFLD development and how? Authors provided link through SCFAs and inflammation is not new and not specific.

We thank the reviewer for this suggestion. We have added the required information.

  1. Figure is not giving any new information and in fact authors never spelled the terms such as oxidative stress and endotoxemia in the content. The authors should list the microbial signatures in IBD and show their contribution to NAFLD development, specifically, rather general.

We thank the reviewer for these suggestions. We added to the figure the required details.

  1. Please provide a table or illustration showing the microbiota signatures in NAFLD, IBD and with combination.

We thank the reviewer for this suggestion. Indeed, we have added to the figure the required signatures of gut dysbiosis.

  1. The authors described the mechanisms involved in the development of IBD with changes in microbiota, similar manner they should provide some information on what pathway they affect within the liver.

We thank the reviewers for these observations. We have provided these information.

  1. How SCFAs contribute to NAFLD development under IBD? The review touched a sentence on the role of SCFAs on lipogenesis on adipose tissue (Line 195-196), but this is not sufficient and does not makes sense.

We thank the reviewer for this observation. We have implemented this sentence.

Again, overall the link between the NAFLD and IBD is missing and therefore provide more information to make the review interesting. 

We thank the reviewers for these observations. We have implemented the text accordingly. In detail, we have added data on gut microbiota metabolites implicated in steatosis pathogenesis in IBD patients.
